# Serum *N*-Glycomics Stratifies Bacteremic Patients Infected with Different Pathogens

**DOI:** 10.3390/jcm10030516

**Published:** 2021-02-01

**Authors:** Sayantani Chatterjee, Rebeca Kawahara, Harry C. Tjondro, David R. Shaw, Marni A. Nenke, David J. Torpy, Morten Thaysen-Andersen

**Affiliations:** 1Department of Molecular Sciences, Macquarie University, Sydney, NSW 2109, Australia; sayantani.chatterjee@hdr.mq.edu.au (S.C.); rebeca.kawaharasakuma@mq.edu.au (R.K.); harry.tjondro@hdr.mq.edu.au (H.C.T.); 2Biomolecular Discovery Research Centre, Macquarie University, Sydney, NSW 2109, Australia; 3Infectious Diseases Clinic, Royal Adelaide Hospital, Adelaide, SA 5000, Australia; david.shaw@sa.gov.au; 4Endocrine and Metabolic Unit, Royal Adelaide Hospital, Adelaide, SA 5000, Australia; marni.nenke@sa.gov.au (M.A.N.); david.torpy@sa.gov.au (D.J.T.); 5School of Medicine, University of Adelaide, Adelaide, SA 5000, Australia; 6Department of Endocrinology and Diabetes, The Queen Elizabeth Hospital, Woodville South, SA 5011, Australia

**Keywords:** bacteremia, biomarker, mass spectrometry, *N*-glycan, *N*-glycomics, porous graphitized carbon, serum

## Abstract

Bacteremia—i.e., the presence of pathogens in the blood stream—is associated with long-term morbidity and is a potential precursor condition to life-threatening sepsis. Timely detection of bacteremia is therefore critical to reduce patient mortality, but existing methods lack precision, speed, and sensitivity to effectively stratify bacteremic patients. Herein, we tested the potential of quantitative serum *N*-glycomics performed using porous graphitized carbon liquid chromatography tandem mass spectrometry to stratify bacteremic patients infected with *Escherichia coli* (*n* = 11), *Staphylococcus aureus* (*n* = 11), *Pseudomonas aeruginosa* (*n* = 5), and *Streptococcus viridans* (*n* = 5) from healthy donors (*n* = 39). In total, 62 *N*-glycan isomers spanning 41 glycan compositions primarily comprising complex-type core fucosylated, bisecting *N*-acetylglucosamine (GlcNAc), and α2,3-/α2,6-sialylated structures were profiled across all samples using label-free quantitation. Excitingly, unsupervised hierarchical clustering and principal component analysis of the serum *N*-glycome data accurately separated the patient groups. *P. aeruginosa*-infected patients displayed prominent *N*-glycome aberrations involving elevated levels of fucosylation and bisecting GlcNAcylation and reduced sialylation relative to other bacteremic patients. Notably, receiver operating characteristic analyses demonstrated that a single *N*-glycan isomer could effectively stratify each of the four bacteremic patient groups from the healthy donors (area under the curve 0.93–1.00). Thus, the serum *N*-glycome represents a new hitherto unexplored class of potential diagnostic markers for bloodstream infections.

## 1. Introduction

Bacteremia, or “blood poisoning”, is the presence of viable pathogenic bacteria in the normally sterile bloodstream [1]. Bacteremia commonly arises from primary bloodstream infections such as intra-vascular catheters, surgery, or untreated urinary tract infections via secondary bloodstream infections that include cuts and wounds on the skin or through bacteria in the nasal airways. While primary bloodstream infections often affect multiple organs, secondary bloodstream infections are generally less fatal [2,3,4].

Gram-negative bacteremia is generally more serious as pathogens leading to this condition are often hospital-derived (e.g., from intensive care units) tend to be more resistant to antibiotics largely because of their impermeable cell wall, and their ability to cause a greater inflammatory response in the host [5,6]. Common bacteria causing gram-negative bacteremia include *Escherichia coli* associated with urinary tract infections [7,8], and *Pseudomonas aeruginosa* and *Klebsiella pneumoniae* commonly associated with lung infections [9,10]. Gram-positive bacteremia, on the other hand, is considered a less harmful condition that most often originates from the bacteria residing on the skin and in the gastrointestinal tract [11]. *Staphylococcus* spp. and *Streptococcus* spp. are the most common skin microbial flora known for their ability to cause gram-positive bacteremia if they successfully enter the bloodstream in sufficient numbers [12].

Bloodstream infections, if undetected or left untreated, may lead to sepsis and septic shock, a serious health condition associated with high mortality and long-term morbidity. The lack of accurate, rapid, and sensitive diagnostics to effectively detect pathogens in blood at relatively low titer is an obstacle that often prevents timely intervention [13,14,15,16]. At present, blood cultures are obtained from infected patients using an intravenous access device. Gram staining is then performed to classify the infecting pathogen. However, this method is slow and suffers from false positive results [17]. Conversely, approximately 50% of patients with life threatening septic shock have negative blood cultures and have a similar illness course to those with positive cultures suggesting false negative cultures in many cases [18]. Other methods to diagnose bacteremia in the clinic include the measurement of white blood cells [19], the neutrophil-to-monocyte ratio [20], or the level of inflammatory markers such as C-reactive protein (CRP) [21], procalcitonin [22], and interleukin-6 [23]. Yet these alternative methods also lack the required precision and sensitivity to identify bacteremic patients at an early stage and are typically unable to stratify bacteremic patients from other systemic inflammatory conditions [24]. Furthermore, many cases of septic shock and other severe infections, such as endocarditis, yield negative blood cultures, sometimes because of empirical antibiotic pre-treatment. Failure to identify an inciting pathogen impairs antimicrobial stewardship and can lead to over- or under-treatment of infections and antibiotic resistance. The development of novel methods for identifying pathogens in culture-negative patients is critical.

Protein *N*-glycosylation, the addition of complex carbohydrates (glycans) to polypeptide chains via sequon-located asparagine residues, plays important roles in diverse inter- and intra-cellular processes [25]. Being a non-template-based and physiology-dependent type of modification, protein *N*-glycosylation is highly dynamic and is known to be altered with a variety of diseases including autoimmune disorders [26,27], cancer [28,29,30,31], and, of importance to this study, inflammation [32,33,34] including peripheral and bloodstream infections [35,36]. Disease-driven aberrations of the *N*-glycome in human plasma and serum have been reviewed [37,38,39].

Protein-bound *N*-glycans are abundantly present in most if not all accessible bodily fluids including in blood and are therefore recognized as attractive yet still clinically under-utilized marker candidates for a variety of pathophysiological conditions including rheumatoid arthritis [40], chronic liver cirrhosis [41], acute pancreatitis [42], urothelial carcinomas [43], and multiple myeloma [44]. Developments in mass spectrometry (MS)-based quantitative *N*-glycomics including the implementation and improvement of porous graphitized carbon liquid chromatography tandem mass spectrometry (PGC-LC-MS/MS)-based methods to quantitatively profile the glycome with glycan isomer resolution [45,46,47,48,49,50,51,52,53] have opened new avenues to test the biomarker potential of glycans for such pathophysiological conditions [28].

The host glycome remains poorly studied in the context of bacteremia. We were only able to find a single study that used targeted *N*-glycoproteomics to investigate the plasma from pathogen-infected patients [54]. In that study, albumin-depleted plasma from febrile patients with (*n* = 9) and without (*n* = 10) blood cultures were enriched for *N*-glycopeptides using lectin chromatography prior to MS detection, which led to the identification of 24 intact *N*-glycopeptides from 8 plasma glycoproteins. The study neither addressed the plasma glycan fine structures associated with bacteremia nor explored the plasma *N*-glycome remodeling arising from bloodstream infections. Thus, it remains unclear if pathogen-mediated changes to the *N*-glycome can be used to stratify bacteremic patients from non-infected individuals.

To this end, we applied quantitative glycomics to sera from a cohort of bacteremic patients and healthy individuals to test the potential of the serum *N*-glycome to inform on the infection status and causative pathogens in patients suffering from bloodstream infection. Our data indicated that serum *N*-glycomics is a hitherto unexplored avenue to diagnose patients suffering from infectious diseases.

## 2. Experimental Section

### 2.1. Materials

#### 2.1.1. Chemicals and Reagents

Ultra-high-quality water came from a Milli-Q system (Merck-Millipore, Melbourne, Australia). Recombinant *Elizabethkingia miricola* peptide-*N*-glycosidase F (PNGase F) produced in *Escherichia coli* was from Promega. Other chemicals, reagents, and proteins were from Sigma-Aldrich (Sydney, Australia) or Thermo Fisher Scientific (Sydney, Australia) unless otherwise specified.

#### 2.1.2. Sample Cohort

Whole blood of healthy donors and bacteremic patients were collected at the Royal Adelaide Hospital (RAH). Ethics approvals were obtained by the RAH Human Research Ethics Committee (HREC/14/RAH/130 and HREC/14/RAH/553, 2018) for the collection and analysis of healthy and bacteremic blood, respectively. Blood was collected from healthy donors from outpatient clinics through community advertising. The donor criteria included no known disease of the hypothalamic-pituitary-adrenal axis, not being pregnant, not taking any contraceptive medicine, not undergoing hormone replacement therapy, and not showing any active inflammatory or infectious conditions.

Bacteremic blood was collected from in-patient admissions at RAH. Samples positive for both gram-negative bacteria, i.e., *Escherichia coli* (*n* = 11) and *Pseudomonas aeruginosa* (*n* = 5), and gram-positive bacteria, i.e., *Staphylococcus aureus* (*n* = 11) and *Streptococcus viridans* (*n* = 5) were included in this study (Figure 1a). These 32 pathogen-positive samples were complemented with an age- and gender-balanced cohort of healthy donors (*n* = 39). See Table 1 and Appendix A for details of the entire sample cohort and clinical data.

### 2.2. Methods

The experimental workflow is depicted in Figure 1b.

#### 2.2.1. Blood Collection

Whole blood from healthy donors and bacteremic patients were collected in serum-separator tubes. After centrifugation, sera were aliquoted and stored at −20 °C until use. The bacteremic and healthy sera were previously studied in another context [55,56]. Aerobic and anaerobic blood culture samples were collected from bacteremic patients and examined by a trained microbiologist who identified the causative organism [55,56]. The total protein concentration of the serum samples was determined using Bradford assays according to the manufacturer’s instructions ahead of the *N*-glycome profiling experiments.

#### 2.2.2. *N*-glycan Release and Preparation

The *N*-glycans were released from the serum proteins and prepared for glycomics, as previously described [45]. No technical replicates (sample handling or LC-MS/MS injection) were performed. In brief, 20 μg total protein from each serum sample was reduced using 10 mM aqueous dithiothreitol (DTT) for 45 min at 56 °C, and carbamidomethylated using 25 mM aqueous iodoacetamide for 30 min in the dark at 20 °C, before the alkylation reaction was quenched using 30 mM aqueous DTT (final concentrations stated).

The proteins were then immobilized on a primed 0.45 μm PVDF membrane (Merck-Millipore). The dried spots were stained with Direct Blue, excised, transferred to separate wells in a flat bottom polypropylene 96-well plate (Corning Life Sciences, Melbourne, Australia), blocked with 1% (*w*/*v*) polyvinylpyrrolidone in 50% (*v*/*v*) aqueous methanol, and washed with water. De-*N*-glycosylation was performed using 2 U PNGase F per 20 μg protein in 10 μL water/well for 16 h at 37 °C. The released *N*-glycans were transferred into fresh tubes and hydroxylated by the addition of 100 mM aqueous ammonium acetate, pH 5 for 1 h at 20 °C. The glycans were reduced using 1 M sodium borohydride in 50 mM aqueous potassium hydroxide for 3 h at 50 °C. The reduction reaction was quenched using glacial acetic acid. Dual desalting of the reduced detached *N*-glycans was performed using, firstly, SCX resin (AG 50W-X8 Resin, Bio-Rad, Sydney, Australia) (where the *N*-glycans are not retained) and then, secondly, PGC (where *N*-glycans are retained) custom-packed as micro-columns on top of C_18_ discs in P10 solid-phase extraction (SPE) formats. The *N*-glycans were eluted from the PGC SPE columns using 0.05% trifluoroacetic acid: 40% acetonitrile (ACN): 59.95% water (*v*/*v*/*v*), dried and reconstituted in 20 μL water, centrifuged at 14,000× *g* for 10 min at 4 °C, and transferred into high recovery glass vials (Waters) for LC-MS/MS analysis. Bovine fetuin was included as a sample handling and LC-MS/MS control.

#### 2.2.3. *N*-glycome Profiling

The *N*-glycome profiling data forming the foundation of this study were acquired using an established PGC-LC-MS/MS method [45,52]. Briefly, the serum *N*-glycans were injected on a heated (50 °C) HyperCarb KAPPA PGC-LC column (particle/pore size, 5 μm/250 Å; column length, 100 mm; inner diameter, 0.18 mm, Thermo Hypersil, Runcorn, UK). The *N*-glycans were separated over an 86 min linear gradient of 0–45% (*v*/*v*) ACN (solvent B) in 10 mM aqueous ammonium bicarbonate (solvent A). A constant flow rate of 3 µL/min was maintained with a post-column make-up flow supplying pure ACN delivered by a Dionex Ultimate-3000 HPLC (Thermo Fisher Scientific, Sydney, Australia). The separated *N*-glycans were ionized using electrospray ionization and detected in negative ion polarity mode using a linear trap quadrupole Velos Pro ion trap mass spectrometer (Thermo Fisher Scientific, Sydney, Australia) with a full scan acquisition range of *m/z* 500–2000, a resolution of *m/z* 0.25 full width half maximum and a source voltage of +3.2 kV. The automatic gain control for the MS1 scans was set to 5 × 10^4^ with a maximum accumulation time of 50 ms. For the MS/MS events, the resolution was set to *m/z* 0.25 full width half maximum, the automatic gain control was 2 × 10^4^ and the maximum accumulation time was 300 ms.

Data-dependent acquisition was enabled for all MS/MS experiments. The three most abundant precursors in each MS1 full scan were selected using resonance activation (ion trap) collision-induced dissociation (CID) at a normalized collision energy of 33%. All MS and MS/MS data were acquired in profile mode and dynamic exclusion was inactivated. The mass accuracy of the precursor and product ions were typically better than 0.2 Da. The LC-MS/MS instrument was tuned and calibrated, and its performance bench marked using bovine fetuin *N*-glycans prior to use. Importantly, the injection order of all samples was randomized to prevent systematic errors in the LC-MS/MS data collection that may otherwise risk masking biological differences and/or introduce artificial differences between patient groups. The robustness, sensitivity, quantitative accuracy and reproducibility of the applied PGC-LC-MS/MS glycan profiling method have previously been documented [52].

The raw data files of all LC-MS/MS datasets included in this study were browsed, interrogated, and annotated using Xcalibur v2.2 (Thermo Fisher Scientific, Sydney, Australia), GlycoMod [57] and GlycoWorkBench v2.1 and via manual de novo glycan sequencing, as previously published [50,58]. Glycans were identified based on the monoisotopic precursor mass, the MS/MS fragmentation pattern, and the relative and absolute PGC-LC retention time of each glycan (see Appendix A for examples of raw spectral data including a typical base peak chromatogram, extracted ion chromatograms featuring LC elution times, isotopic envelopes and signal-to-noise ratios of glycans of high importance to the findings of this work, and MS/MS spectral evidence of all reported glycans). The relative abundances of the individual *N*-glycans were determined from area under the curve (AUC) measurements based on extracted ion chromatograms performed for all relevant charge states of the monoisotopic precursor *m/z* using RawMeat v2.1 (Vast Scientific) and Skyline (64-bit) v20.1.0.76 [46]. Extremely low abundant *N*-glycans with a relative abundance less than 0.01% were excluded from the quantitation due to poor signal-to-noise ratios within the resulting MS and MS/MS spectra. Examples of EICs of several low abundance glycans included in the quantitative analysis (0.01–0.15% relative abundance of the total glycome) have been provided in Appendix A.

#### 2.2.4. Statistics

The quantitative glycomics data were subjected to different statistical tests performed without data transformation and were visualized in various ways with or without transformation of the relative abundance data. Unpaired two-tailed Student’s *t*-tests were performed where *p* < 0.05 was chosen as the confidence threshold. Statistical confidence has been indicated by * (*p* < 0.05), ** (*p* < 0.01), *** (*p* < 0.001), and **** (*p* < 0.0001). NS was used to indicate that no significance was found (*p* ≥ 0.05). Multiple one-way sample analysis of variance (ANOVA) tests were also applied. For the ANOVA tests, the significance threshold was *p* < 0.05 after Benjamini–Hochberg false discovery rate (FDR) correction. Post-hoc analysis was performed using Fisher’s least significance difference (LSD) test and the significance threshold was *p* < 0.05. Data points have been plotted as the mean and error bars represent their standard deviation (SD). Heat-maps and hierarchical clustering analyses were performed with Perseus v1.6.7 using Euclidean distance with average linkages. The relative abundance values of the glycans were used as input data for these analyses after log2 or Z-score transformation. Z-score transformation, a data transformation strategy commonly used in proteomics and glycomics to normalize data according to the observed mean and standard deviation across sample cohorts [59], was performed to better compare and visualize the changes across conditions. Receiver operating characteristic (ROC) curves of both univariate and multivariate analyses were performed using linear support vector machines classification methods. The confidence threshold was AUC > 0.75 with 1.00 representing perfect separation. For the ROC curves and principal component analysis (PCA) that were performed using MetaboAnalyst [60], untransformed relative abundance values of the glycans identified and quantified across all biological replicates from each patient group were used as input data.

## 3. Results

Whole (non-depleted) sera from 39 healthy donors and 32 bacteremic patients were profiled using quantitative glycomics, as shown in Figure 1a. The healthy cohort comprised 19 females and 20 males (18–80 years). The bacteremic cohort consisted of 11 *E. coli-*, 11 *S. aureus*-, 5 *P. aeruginosa*-, and 5 *S. viridans*-infected individuals covering both female and male subjects (21–90 years). PGC-LC-MS/MS, a widely used method in quantitative glycomics [28,46,47,52,58], was used to quantitatively profile the fine structures of all *N*-glycan isomers released from the serum proteins including their monosaccharide compositions, their topology/branch patterns, and key glycosidic linkages, as shown in Figure 1b.

### 3.1. Quantitative N-glycome Profiling

A total of 62 *N*-glycan isomers (herein denoted glycan 1-41c) spanning 41 different glycan compositions were manually identified from the investigated samples, as shown in Figure 2a (see Appendix A for examples of spectral raw data and fully annotated MS/MS spectral evidence supporting all reported *N*-glycans). As is common practice in comparative glycomics [28,29,61], the relative abundance of all 62 *N*-glycan isomers was determined within each sample using label-free quantitation and the resulting glycan profiles were then compared across all samples, as shown in Appendix A. In agreement with literature of the human serum *N*-glycome [41,42,43,44], the *N*-glycome data for all individuals were found to span the three common *N*-glycan classes including, most prominently, the complex-type *N*-glycans identified with and without sialylation and fucosylation (~94.5%), and several less abundant oligomannosidic- (~4.0%), and hybrid- (~1.5%) type *N*-glycans, as shown in Figure 2b.

The distribution of the glycan classes and their structural features including the number of antennas, presence of bisecting GlcNAc, and the positions and linkages of the terminal sialic acid and fucose residues were statistically compared across all samples using ANOVA followed by Fisher’s LSD test, as shown in Figure 2c. Interestingly, all the glycan classes and structural features showed significant differences between the bacteremic patient groups and the healthy donors indicating considerable pathogen-mediated changes to the serum *N*-glycome. In particular, the level of core (α1,6-) fucosylation (*p* < 0.05, *P. aeruginosa—E. coli; P. aeruginosa*—Healthy; Healthy—*S. viridans; P. aeruginosa—S. aureus; P. aeruginosa—S. viridans; S. aureus—S. viridans*) and α2,6-sialylation (*p* < 0.05, *E. coli*—Healthy; *E. coli—P. aeruginosa; E. coli—S. aureus;* Healthy—*P. aeruginosa; S. viridans*—Healthy; *S. aureus—P. aeruginosa; S. viridans—P. aeruginosa; S. viridans—S. aureus*) differed strongly across the patient groups and the healthy cohort. Notably, the *P. aeruginosa*-infected individuals displayed relative high levels of core fucosylation (37.3 ± 9.7%, *p* = 5.9 × 10^−10^) and low levels of α2,6-sialylation (59.3% ± 6.4%, *p* = 4.0 × 10^−8^) relative to all other bacteremic patients and healthy donors (core fucose range 15.9–22.8% and α2,6-sialylation range 63.5–72.2%, respectively), as shown in Appendix A.

### 3.2. Serum N-glycomics Separates Bacteremic Patients from Healthy Donors without Prior Knowledge

To explore the relationship between the serum *N*-glycome established for the 71 individuals included in this study, we first performed unsupervised hierarchical clustering and heat-map analyses using the relative abundances of all identified and quantified glycan isomers (glycan 1-41c) after log2 transformation, as shown in Figure 3.

At a glance, the resulting heat-map appeared surprisingly uniform across the 71 samples. In line with published *N*-glycome profile data of human serum [41,42,43,44], the mono- and di-sialylated biantennary complex-type *N*-glycans (glycan 21 and 23a, respectively) were found to be abundant structures collectively accounting for more than half of the *N*-glycome across all samples. Further, inspection of the heat-map illustrated that other glycan isomers, detected at lower levels (0.01–5%), were also rather uniformly expressed with only relatively subtle differences observed across the 71 samples. Some very low abundant glycan isomers were not detected in a few samples (indicated as grey pixels in the heat-map), due to minor technical variations between the individual MS/MS runs and/or due to biological variation within each patient group.

Excitingly, the unsupervised hierarchical cluster analysis effectively separated each of the four bacteremic patient groups from the healthy donors. As noted above, the *P. aeruginosa*-infected patients displayed the most aberrant serum *N*-glycome relative to the other subjects as illustrated by a clear segregation of these individuals within a separate cluster. A few pathogen-infected individuals, such as *P. aeruginosa* 12 and *E. coli* 7, and *E. coli* 11 clustered with the healthy donors rather than with their respective patient groups. Interrogation of the available clinical and patient meta-data did not reveal any confounding factors or associations to the *N*-glycome profile data that could explain the less-than-perfect clustering of these few individuals. The separation of most, but not all, individuals in each patient group indicated that the serum *N*-glycome underwent subtle yet consistent remodeling upon pathogen infection and suggests that such glycome changes may be useful to identify infected individuals and the causative pathogens from their healthy counterparts without prior knowledge of their health status.

ROC analyses were then performed to explore the potential of specific *N*-glycan isomers to stratify bacteremic patients from healthy donors. Notably, these analyses showed that a single *N*-glycan isomer can accurately stratify each of the four patient groups from the healthy individuals with high confidence. Specifically, a hybrid-type asialoglycan (glycan 9) was able to stratify *E. coli*-infected individuals from the healthy donors (AUC = 0.930, *p* = 1.7 × 10^−7^), as shown in Figure 4a. In addition, a GlcNAc-capped monoantennary core fucosylated *N*-glycan (glycan 11) perfectly stratified *P. aeruginosa*-infected individuals from the healthy donor cohort (AUC = 1.000, *p* = 6.7 × 10^−11^), Figure 4b. Further, an α2,6-sialylated isomer of the rather unusual monoantennary sialoglycan (glycan 12a) showed a clear potential for stratifying *S. aureus*-infected individuals from healthy donors (AUC = 0.991, *p* = 2.7 × 10^−13^), as shown in Figure 4c. Finally, a α2,6-sialylated triantennary *N*-glycan isomer carrying antenna fucosylation (glycan 40a) perfectly separated *S. viridans*-infected individuals from the healthy donor cohort (AUC = 1.000, *p* = 8.2 × 10^−12^), as shown in Figure 4d. EICs and mass spectral data showing the PGC-LC elution times and isotopic envelope of these four critical *N*-glycans are presented in Appendix A.

### 3.3. Pathogen-Specific Alterations of the Serum N-glycome

In the search for pathogen-specific glycosignatures, a heat-map, and cluster analysis were performed for a panel of 17 glycan isomers that were differentially expressed in serum across the bacteremic patient groups (ANOVA, *p* < 0.05), Figure 5a. Healthy donor data were not included in this specific analysis. Amongst this panel of *N*-glycans, 11 structures comprising mostly fucosylated and bisecting GlcNAcylated *N*-glycans were significantly elevated in sera from *P. aeruginosa*-infected individuals (red color coding) relative to the other bacteremic patient groups. Unsupervised PCA using the same panel of 17 glycan isomers as input demonstrated a complete segregation of the *P. aeruginosa*-infected individuals from the other bacteremic patient groups that remained partially unseparated, Figure 5b. The aberrant serum glycosylation of the *P. aeruginosa*-infected individuals was supported by a clear elevation of fucosylation (*p* < 0.0001) and bisecting GlcNAcylation (*p* < 0.01 for *P. aeruginosa*-infected individuals *versus* all other patients) and reduction of sialylation (*p* < 0.0001 for *P. aeruginosa*-infected individuals *versus* all other patient groups except for *S. aureus*-infected individuals) compared to other bacteremic patients, Figure 5c–e. EICs and mass spectral data showing the PGC-LC elution times and isotopic envelope of the 11 aberrant *N*-glycans in sera from *P. aeruginosa*-infected individuals are presented in Appendix A.

## 4. Discussion

We are the first to use quantitative glycomics to establish the *N*-glycome of sera from four groups of bacteremic patients infected with either *P. aeruginosa*, *E. coli*, *S. aureus*, or *S. viridans,* as well as a healthy donor cohort. PGC-LC-MS/MS profiling enabled us to accurately quantify the *N*-glycome differences between bacteremic and healthy sera with glycan isomer resolution. Many pathogen-specific alterations to the *N*-glycome were observed including quantitative changes in the distribution of glycan types, e.g., oligomannosylation and changes in the level of key structural features of the complex-type *N*-glycans, e.g., bisecting GlcNAcylation, α2,6-, and α2,3-sialylation, as well as core and antenna fucosylation.

Excitingly, the serum *N*-glycome showed a hitherto unknown potential to stratify bacteremic patients from healthy donors, and even an ability to segregate individuals infected with different pathogens, as supported by three independent statistical methods. Unsupervised hierarchical clustering and PCA demonstrated that accurate stratification of the bacteremic patients can be achieved using the entire serum *N*-glycome and a panel of 17 *N*-glycan isomers as input, respectively. The *P. aeruginosa*-infected individuals were particularly well stratified from other patient groups. Further, ROC analyses showed that four structurally and biosynthetically unrelated glycan isomers were able to effectively stratify bacteremic patients infected with different pathogens. Specifically, a hybrid-type asialoglycan (glycan 9) stratified *E. coli-*, a β1,2-GlcNAc-capped monoantennary core fucosylated *N*-glycan (glycan 11) stratified *P. aeruginosa-*, an α2,6-sialylated monoantennary *N*-glycan (glycan 12a) stratified *S. aureus-*, and an α2,6-sialylated triantennary antenna fucosylated *N*-glycan (glycan 40a) stratified *S. viridans*-infected individuals from healthy donors. This is, to the best of our knowledge, the first report documenting that the serum *N*-glycome and parts thereof can be utilized to diagnose bloodstream infections with precision.

Amongst the four causative bacteremic pathogens explored in this study, *P. aeruginosa*-infected individuals showed the most aberrant *N*-glycome profiles relative to other patient groups. Sera from *P. aeruginosa*-infected patients displayed elevated levels of bisecting GlcNAcylation and core fucosylation, and a concomitant decrease in sialylation. Being an opportunistic gram-negative pathogen, *P. aeruginosa* is closely associated with cystic fibrosis [63,64], affecting lung and liver [65,66], and has been reported to alter the host glycome by modulation of the complex- and paucimannosidic-type *N*-glycans (discussed below) in peripheral tissues and in blood [54,67]. Albeit less studied, *P. aeruginosa*-based bloodstream infections have been reported to lead to an elevation of circulating immunoglobulins [68] and acute-phase glycoproteins such as α-1-antitrypsin, fibrinogen, and haptoglobin [69]. The elevation of immunoglobulin is particularly interesting as it may, in part, explain the glycome remodeling observed in *P. aeruginosa*-infected sera, i.e., elevation of core fucosylation and bisecting GlcNAcylation, and reduced sialylation. Asialylated *N*-glycans carrying core fucosylated and bisecting GlcNAc are namely structures commonly associated with immunoglobulin G [70,71]. Further to this, *P. aeruginosa* is also known to express a sialidase that reportedly cleaves sialic acid residues from host glycoproteins [72], and is capable of absorbing host α2,6- and α2,3- sialoglycoproteins [73]. These two mechanisms may also contribute to the sialic acid-poor glycophenotype of *P. aeruginosa*-infected individuals.

Despite the presence of several *E. coli-*, *S. aureus-,* and *S. viridans*-associated glycome signatures evidently useful for the stratification of bacteremic patients infected with such gram-positive and negative pathogens, no structural commonalities, biosynthetic mechanisms, or glycome remodeling processes could be identified to explain the quantitative serum *N*-glycome alterations observed for these pathogens. Thus, further research is required to decipher the molecular basis of how these three pathogens alter the host serum glycome.

The discovery of the serum *N*-glycome as a molecular reporter of bloodstream infections was driven by the accurate quantitation of glycan isomers enabled by the PGC-LC-MS/MS profiling method. PGC-LC-MS/MS is considered one of the gold standards in glycomics [45,47,48,49,74,75], but remains restricted to relatively few glycomics laboratories across the world and is still less streamlined than the more conventional LC-MS/MS approaches utilized in proteomics and may thus not be an immediately useful technique in the clinic. Antibodies or lectins with affinities to bisecting GlcNAc and/or linkage-specific sialylation or fucosylation represent alternative methods that may display a greater potential for direct clinical implementation in the context of bacteremia.

Interrogation of the *N*-glycome data from the 39 healthy donors did not reveal any significant age- or gender-related glycome features. Thus, although age and gender have previously been reported to impact (in relatively subtle ways) the serum *N*-glycome and IgG *N*-glycosylation [76,77,78], we did not consider these variables to be confounding factors in our study. In fact, we were unable to identify using correlation analyses any confounding factors within the extensive clinical meta-data available for the studied bacteremic patient cohort that may contribute to the intriguing pathogen-mediated glycome changes observed herein (e.g., body mass index, blood pressure, smoking habits, antibiotic intake, Pitt score, CRP levels, and neutrophil/white blood cell counts, as shown in Table 1 and Appendix A). Furthermore, the simultaneously processing of all samples and their randomized injection order on the mass spectrometer strongly suggest that the observed glycome differences were not of technical origin. Finally, as a test for repeatability, all serum samples were processed and analyzed multiple times on different days (months apart) by two different analysts only to show the same stratification profile of the serum samples.

As expected, *N*-glycans belonging to the three common *N*-glycan types—i.e., oligomannosidic, hybrid, and complex types—were observed in both the healthy and bacteremic sera [79]. A fourth less explored class of human *N*-glycans, namely of the paucimannosidic type [80], were not observed in any of the investigated samples. Paucimannosidic *N*-glycans is a type of immuno-modulatory *N*-glycosylation comprising truncated structures (Man_1–3_GlcNAc_2_Fuc_0–1_) expressed and stored by resting neutrophils and monocytes in blood circulation [28,29,32,33,61,81,82]. While the absence of paucimannosidic *N*-glycans in the healthy sera was expected based on existing literature of human serum *N*-glycosylation [79,83,84], the absence of paucimannosidic *N*-glycans in bacteremic sera was surprising given the documented degranulation of paucimannosidic proteins from pathogen-activated blood neutrophils [67,82], as well as the fact that our clinical data indicated systemic inflammation presumably involving neutrophil activation (e.g., high neutrophil counts and CRP levels, see Table 1). The lack of paucimannosidic *N*-glycans in bacteremic sera may be attributed to the dramatic dilution effects in and/or rapid removal of these neutrophil glycans from blood circulation or may alternatively be a consequence of an only incomplete neutrophil activation/degranulation at the pathogen titer level experienced by bacteremic patients, aspects we are currently exploring.

In conclusion, this study has shown that the serum *N*-glycome represents a new hitherto unexplored class of potential diagnostic markers for bloodstream infections. The ability to stratify bacteremic patients with relatively low Pitt scores before they become critically ill and/or enter septic shock is particularly exciting since early diagnosis of bacteremia and indeed identification of the causative pathogen may provide better opportunities for clinical intervention with better patient outcome. The *N*-glycan-centric findings reported herein may be extended in the future to also include information of the protein carriers of glycosylation using quantitative glycoproteomics methods that are becoming increasingly accessible in the field [29,85,86,87]. Such future studies are likely to advance our glycobiological knowledge of how pathogens alter the molecular makeup of the host and conversely how the host responds to pathogen infection, efforts that may eventually lead to robust glycosylation-based diagnostic markers and therapeutics for bacteremia.

## Figures and Tables

**Figure 1 jcm-10-00516-f001:**
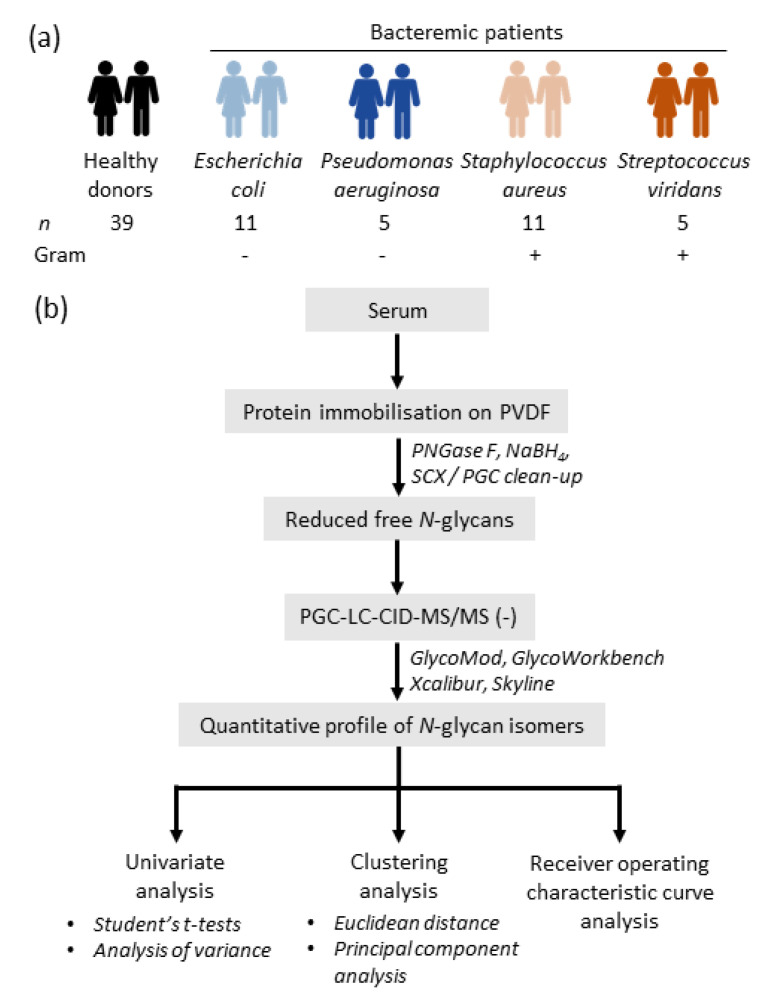
Study design. Overview of the (**a**) investigated cohort comprising four groups of bacteremic patients infected with different pathogens and a healthy control group and (**b**) the experimental workflow. Quantitative *N*-glycome profiling was performed of the bacteremic and healthy donor sera using an established glycomics method [52] and statistical analyses were applied to the glycomics data. NaBH_4_, sodium borohydride; PGC-LC-CID-MS/MS, porous graphitized carbon liquid chromatography collision-induced-dissociation tandem mass spectrometry; PNGase F, peptide-*N*-glycosidase F; PVDF, polyvinylidene fluoride; SCX, strong cationic exchange.

**Figure 2 jcm-10-00516-f002:**
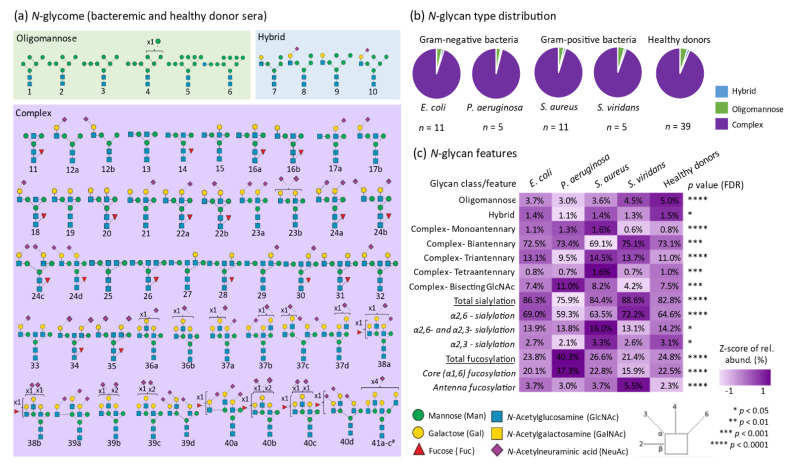
Overview of the *N*-glycome profile of bacteremic and healthy donor sera. (**a**) Map of the identified *N*-glycan isomers and the glycan identifiers used in this study (glycan 1-41c). ^#^ a single glycan isomer has been depicted since the exact glycan fine structures could not be determined for these three isomeric structures. Distribution of the (**b**) *N*-glycan types, i.e., complex-, hybrid-, and oligomannosidic-type, and (**c**) structural features of the complex-type *N*-glycans including the antennary patterns, bisecting GlcNAcylation, and the positions and linkages of terminal sialic acid and fucose residues. The mean relative abundances of various glycan features are indicated for all patient groups and the healthy donors and complemented with a heat-map representation after Z- score transformation of the relative glycan abundances. Statistical significance was tested using ANOVA followed by FDR correction (*p* < 0.05). See key for the used symbol and linkage nomenclature [62].

**Figure 3 jcm-10-00516-f003:**
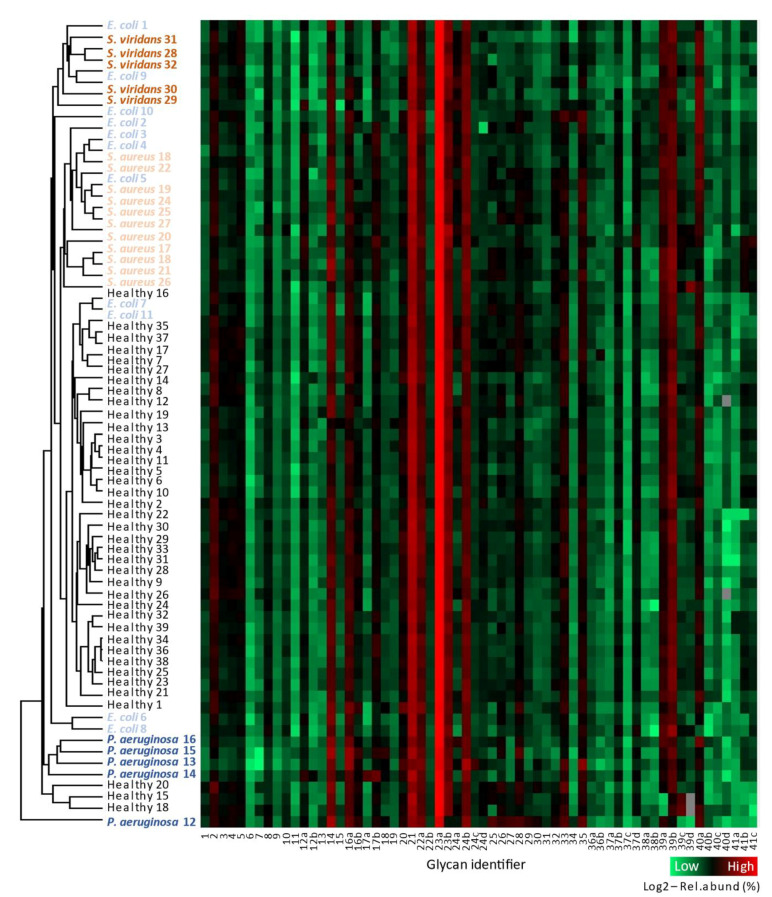
The serum *N*-glycome separates the bacteremic patient groups and healthy donors. Unsupervised hierarchical cluster analysis using Euclidean distance and average linkage (**left**) and heat-map representation of the *N*-glycome profiling data (**right**) performed using the Perseus software after log2 transformation of the relative abundances of the 62 *N*-glycan structures (glycan identifier 1-41c) identified and quantified across all 71 samples investigated in this study. Some very low abundant glycan isomers were not detected in a few samples (indicated as grey pixels in the heat-map), due to minor technical variations between the individual MS/MS runs and/or due to the inherent biological variation within each patient group. See Appendix A for details of the quantitative glycome data. See Figure 2 for structures and glycan identifiers. See Appendix A for metadata and patient identifiers. See Appendix A for spectral evidence.

**Figure 4 jcm-10-00516-f004:**
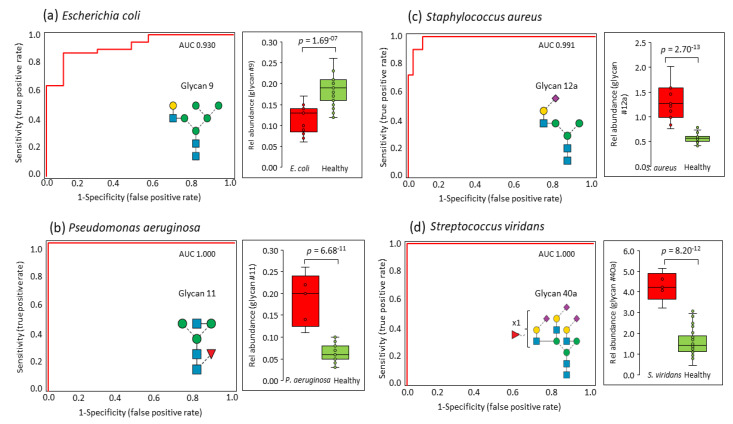
Receiver operating characteristic (ROC) analysis (red trace) of each of the four bacteremic patient groups *versus* the healthy donor group. Notably, a single *N*-glycan isomer was able to accurately stratify patients from healthy donors i.e., (**a**) glycan 9 for *E. coli*-infected individuals, (**b**) glycan 11 for *P. aeruginosa*-infected individuals, (**c**) glycan 12a for *S. aureus*-infected individuals, and (**d**) glycan 40a for *S. viridans*-infected individuals. AUCs and *p* values are shown.

**Figure 5 jcm-10-00516-f005:**
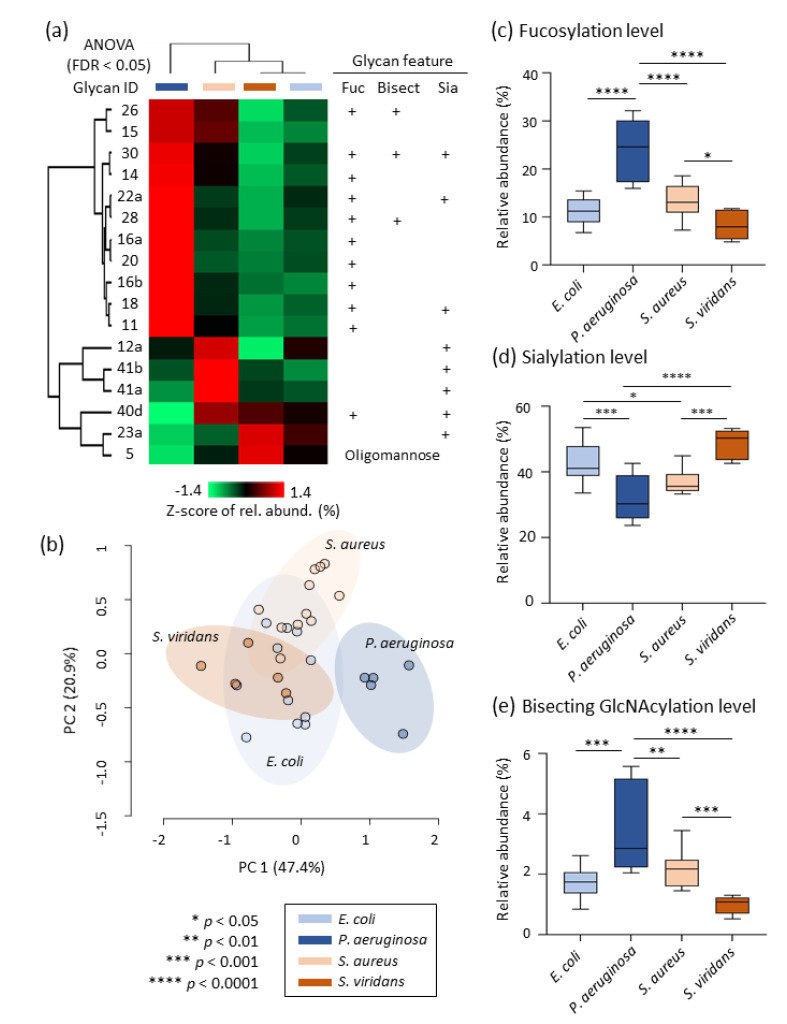
Pathogen-specific alterations of the serum *N*-glycome. (**a**) Heat-map representation illustrating the quantitative profile of selected glycans after Z-score transformation of the relative abundance data and cluster analysis using Euclidean distance and average linkage performed with Perseus. The selected glycans were a panel of 17 glycan isomers that were found to be differentially abundant in serum across the bacteremic patient groups (ANOVA followed by FDR correction, *p* < 0.05). Key structural features are indicated for these glycans. The *P. aeruginosa*-infected individuals displayed a starkly different serum *N*-glycome profile relative to other bacteremic patients. (**b**) Unsupervised PCA performed with the Metaboanalyst software using untransformed relative abundance data of the panel of 17 glycan isomers as input demonstrating a complete separation of *P. aeruginosa*-infected individuals from the other bacteremic patient groups. The *P. aeruginosa*-infected individuals displayed aberrant levels of (**c**) fucosylation, (**d**) sialylation, and (**e**) bisecting GlcNAcylation (ANOVA and Fisher’s LSD, *p* < 0.05).

**Table 1 jcm-10-00516-t001:** Clinical data of the individuals investigated in this study. Data are represented as mean ± SD. N/A, data not available. * Indicates that these values were sourced from published data documenting the normal range of total white cells and neutrophils [55,56] rather than being measured directly from the healthy donors investigated in this study.

*n*	Pathogen (Gram Character)	Age (Years)	Sex (F, Female; M, Male)	Pitt Severity Score	White Cell Count (10^9^/L)	Neutrophil Count (10^9^/L)	C-Reactive Protein (mg/L)
39	Healthy donors	54.9 ± 17.5	F = 19 M = 20	N/A	4.0–11.0 *	2.0–8.0 *	N/A
11	*E. coli* (negative)	61.4 ± 24.8	F = 9 M = 2	1.1 ± 1.0	18.5 ± 11.3	16.7 ± 10.1	186.7 ± 89.8
5	*P. aeruginosa* (negative)	73.2 ± 4.4	F = 0 M = 5	2.0 ± 0.0	7.8 ± 9.9	6.7 ± 9.5	130.4 ± 49.3
11	*S. aureus* (positive)	49.1 ± 18.8	F = 5 M = 6	0.9 ± 1.2	12.6 ± 3.6	10.7 ± 3.5	161.0 ± 123.3
5	*S. viridans* (positive)	43.4 ± 17.5	F = 2 M = 3	0.4 ± 0.5	8.1 ± 7.4	11.2 ± 2.2	127.0 ± 68.7

## Data Availability

All PGC-LC-MS/MS raw data have been made publicly available via the GlycoPOST repository (identifier: GPST000157). The files can be downloaded free of charge from the link https://glycopost.glycosmos.org/preview/10430894245fd6d57d72169 (PIN: 9551).

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
