# Peer review of "Serum N-Glycomics Stratifies Bacteremic Patients Infected with Different Pathogens"

_jcm, 2021, doi:10.3390/jcm10030516_

Round 1
Reviewer 1 Report
In the present manuscript Chatterjee, et al. report on the analysis of PNGase F-released (N-)glycan structures, derived from a cohort of bacteremic patient blood sera and healthy controls. Using conventional, well-estabished LC-MS(/MS)-based methodology for the quantitiation of 62 different glycan structures, the authors show that -indeed- various bacterial infections in the blood-stream seem to cause a limited number of robust alterations of select serum-derived glycan species. No biosynthetic mechanisms or glycome remodelling processes, which would explain their observations, were deduced. The analytical framework remains restricted to dedicated glycomics laboratories and is thus not an immediately useful routine analytical technique in the clinic. The paper is a well written case-study report, but remains superficial.
Major point:
- Applicability of the present quantitative data-set is unclear. Unfortunately, the authors merely conclude (line 532-540) that "the simultaneously processing of all samples and their randomised injection order on the mass spectrometer strongly suggest that the observed glycome differences were not of technical origin. Finally, as a test for repeatability, all serum samples were processed and analysed multiple times on different days (months apart) and by two different analysts only to show the same stratification profile of the serum samples (data not shown)".
To provide the reader with a clearer estimate on the reproducibility of their analyses -particularly in the light of quantifying such extremely low abundant glycan species within the dynamic range of clinical human serum samples- I strongly recommend that the authors provide said supplementary data, or -at least- provide analytical key figures and parameters (e.g. standard-deviations calaculate from of technical triplicates, limit of detection, limit of quantitation, signal-to-noise ratios) in the manuscript. Also, an examplary total ion- or base peak-chromatogram of the analysis, indicating the elution time-points of the most critical glycan structures, would be useful for the non-PGC-LC-MS(/MS)-expert reader.
- Throughout the manuscript, the authors use various advanced statistical operations and tools. Often it is unclear which data exactly were used, or which operations/transformations were performed. I strongly recommend to either extend and detail critical steps in the statistical methods section, or to be more verbose in the respective results sections.
Minor points:
- Fig.2 (c): "% Z-Score" - please explain in more detail what "[...] a heat map representation after Z score transformation of the glycan abundances" means.
-Fig.3: The hierachical clustering is extremely hard to read! Please expand the dendrogram at the expense of the less informative heat-map. Also, a discussion on the occassional clustering of healthy donors with bactericemic samples is warranted: What went wrong with sample "P.aeruginosa 12"? Why do "E.coli 7" and "E.coli 11" cluster with the healthy controls?
In the heat-map: what do the grey bits mean? not detected (and why so)?
-Fig.5 (a): "% Z-Score" - please explain.
-Fig.5 (b): Prinicple component analysis: where is the healthy control group?
-Lines 541-558: Why paucimannosidic N-glycans? it is unclear if paucimannosidic N-glycan structures are detected healthy human serum samples?
Author Response
Please find enclosed the response to the excellent suggestions and comments from this reviewer.

Reviewer 2 Report
In this work, the authors described the potential of quantitative serum N-glycomics performed using PGC LC-MS/MS to stratify bacteremic patients infected with Escherichia coli, Staphylococcus aureus, Pseudomonas aeruginosa, and Streptococcus viridans from healthy donors. The research is well-designed and the methods are adequately described. The results from the work are clearly analyzed and presented, which support the conclusion. Overall, it is a nice work and should be suitable for publication after minor revision.
For example, delete the ; in line 172.
It would be better to shorten and rephrase the sentences from line 228 to 234.
To further improve the manuscript, I have one more suggestion: Could the authors supply a summary table for all identified glycans in the supplementary data, including the structure, compositions, retention time, observed ions, theoretical m/z and mass deviation?Author Response
Please find enclosed the response to the excellent suggestions and comments from this reviewer.

Round 2
Reviewer 1 Report
In the revised version of their manuscript the authors seem to have adressed all my points. Unfortunately, the Supplementary Data section has not been updated accordingly. I can therefore not comment on any additional data the authors may have provided.
Author Response
Reviewer 1: "Unfortunately, the Supplementary Data section has not been updated accordingly. I can therefore not comment on any additional data the authors may have provided."
We apologies if the reviewer did not receive our two revised SI files which we believe we submitted with a track changes and a clean version of the revised manuscript in a zipped file. The two revised SI files were entitled: "Chatterjee_Bacteremia_JClinMed_Revised_Supplementary Data S1-S3.PDF" and "Chatterjee_Bacteremia_JClinMed_Revised_Supplementary Table S1-S3.XLSX".
We will now attempt to upload these two files again via the online portal and also provide the files directly to the manuscript handling editor in case the files do not upload correctly.
In short, the supplementary data file has been updated to include examples of raw mass spectral data as suggested by Reviewer 1 (herein now called Supplementary Data S1-S2, the original MS/MS spectral evidence has been called Supplementary Data S3). Further, the supplementary table S2 has been updated to include mass deviation information (column F-H), retention time information (column Q-U) and AUC values (column W-CP) of the entire set of 62 N-glycans as suggested by the reviewer.
We hope that Reviewer 1 is now satisfied with the manuscript.